# A deep dive into the use of local positioning system in professional handball: Automatic detection of players' orientation, position and game phases to analyse specific physical demands

Thomas Lefèvre[1], Brice Guignard[1]*, Claude Karcher[2,3,4], Xavier Reche[5], Roger Font[5,6,7], John Komar[8]

**1** Univ Rouen Normandie, Normandie Univ, CETAPS UR 3832, Rouen, France, **2** Faculty of Medicine, Mitochondria, Oxidative Stress and Muscular Protection Laboratory (EA 3072), University of Strasbourg, Strasbourg, France, **3** Faculty of Sport Sciences, European Centre for Education, Research and Innovation in Exercise Physiology (CEERIPE), University of Strasbourg, Strasbourg, France, **4** Centre de Ressources, d'Expertises et de Performances Sportives, CREPS de Strasbourg, Strasbourg, France, **5** Barça Innovation Hub, FC Barcelona, Barcelona, Spain, **6** Research Group in Tecnologia Aplicada a l'Alt Rendiment i la Salut (TAARS), Tecnocampus, Department of Health Sciences, Pompeu Fabra University, Mataró, Spain, **7** GRCE Research Group, National Institute of Physical Education of Catalonia (INEFC), Barcelona, Spain, **8** Physical Education and Sports Sciences, National Institute of Education, Nanyang Technological University, Singapore, Singapore

* brice.guignard@univ-rouen.fr

## Abstract

The objective of this study is to automate and analyse the quantification of external load during an elite men's handball match. This study was carried out using data from a local positioning system and inertial measurement units. The literature review leads us to assume that physical demands are different depending on position, player specialty and phases of the game. In order to do this analysis, raw data was used from professional competitors of a Spanish club during National and European competition matches. First, a game phase algorithm was designed to automate phase recognition. Then, a descriptive evaluation of the means and standard deviation was performed with the following variables: total distance, total time, total Accel'Rate, the percentages of distance and time per speed and displacement direction. A Kruskal Wallis test was applied to normalized distance and normalized Accel'Rate. Defensive play showed the highest values on covered distance (930.6 ± 395.0 m). However, normalized distance showed significant differences (p<0.05) across all phases with defensive play (558.8 ± 53.9 m/10min) lower than offensive play (870.3 ± 145.7 m/10min), offensive transition (1671.3 ± 242.0 m/10min) or defensive transition (1604.5 ± 242.0 m/10min). Regarding position, wing players covered the most distance (2925.8 ± 998.8 m) at the second highest intensity (911.4 ± 63.3 m/10min) after offensive back players (1105.0 ± 84.9 m/10min). Significant difference in normalized requirements were found between each playing position: goalkeepers, wings, versatile backs, versatile line players, offensive backs and defensive backs (p<0.05), so a separation between offensive or defensive specialists is plausible and necessary. In conclusion, as physical demands differ for

**Data Availability Statement:** All relevant data for this study are publicly available from an external repository (https://doi.org/10.25340/R4/AIRWI0).

**Funding:** The authors (BG) received funding for this work from CETAPS laboratory (Centre d'études des transformations des activités physiques et sportives) UR 3832. https://cetaps.univ-rouen.fr/ The funders had no role in study design, data collection and analysis, decision to publish, or preparation of the manuscript.

**Competing interests:** The authors have declared that no competing interests exist.

each game phase, activity profile among players is modulated by their playing position and their specialty (offense, defense or none). This study may help to create individual training programs according to precise on-court demands.

## Introduction

In sports, multiple variables may affect athlete's performance, and physical attributes are with no doubt one of the major factors. Nowadays, one of the goals of physical trainers is to achieve a workload high enough to improve the physical qualities of athletes while managing fatigue appearance that can lead to injury [1]. Faced with this dilemma, understanding competition demands became a major topic to determine the precise needs of a team sport. In the past decades physical data in outdoor sports [1, 2] have been collected through global positioning systems (GPS) that capture position as a function of time and inertial measurements units (IMUs, capturing linear acceleration and angular velocity) to determine the external physical load of trainings or games. However, in handball, due to GPS being non applicable for indoor activities, video-based motion analyses or local positioning systems (LPS) are now developing to obtain motion and external load data.

Handball is an Olympic team sport characterized by intermittent high intensity actions like accelerations, decelerations, jumps, changes of direction or collisions. Two teams of seven players (six players and one goalkeeper) compete during 2 periods of 30 min with the objective of creating spaces within the defensive plays to score goals in advantageous conditions (i.e., take a shot in a condition where it is more likely to be scored). Distance covered by elite handball players in game (excluding Goalkeepers) varies from 2757 to 4964 m per game [3]. This distance is split between periods of high intensity actions like side steps, backward running, high-speed running (HSR) or sprints and periods of active recuperation by standing, walking or sitting on the bench [4, 5]. In addition, in-game demands are highly influenced by playing roles and game phases. Previous investigation [5] found that line players spent 1.7% of match time at high intensities, whereas it was 2.2% for back players and 3.6% for wing players. Also, back players covered higher distance per game 4964 ± 642 m than wing players (4234 ± 520 m) or line players (3910 ± 507 m). It has also been shown [5] that the average total distance covered in offensive phases was similar to the one covered in defensive phases (1846 vs 1781 m), but the average speed was lower in offensive phases as compared to defensive phases (6.08 vs 6.75 km·h$^{-1}$). To our knowledge, all those studies did not separate stabilized offensive and defensive phases from offensive and defensive transition phases. Indeed, the transition phases and the stabilized phases are usually merged, showing a lot of high-speed running and sprints, which is most likely due to the activity during the transition phases rather than the stabilized phases.

Most of previous studies had been conducted through video-based motion analyses that are time consuming, however recent publications in handball are using LPS or IMUs [6–8]. This shift in technology reduces computation time as it allows automatic tagging. For instance LPS has been used to analyse match requests of the 2019 Champions League Handball Final Four [6]. A distinction by position was possible with a higher running pace for center and left back in offensive play and for mid-left; front center defender and outside right players in defensive play. There is also a difference between offensive phases and defensive phases (i.e. specialty). Results showed a significant increase in the distance covered in walking and in high intensity running (+20% and +25.2%) and a decrease in the distance covered in jogging (-29.6%) during defensive phases compared to offensive phases. Some studies on women's handball looking at the quantification of high intensity events using IMUs have been reported. Two of those

studies [7, 8] collected data on 20 players belonging to the Norwegian women's national team during international matches. The aim was to quantify high-intensity match events and handball activity profiles using IMUs. Findings showed that high-intensity activities are position-specific and that their intensity depends on the role of the player in the team (offensive specialist, defensive specialist, or player playing in both situations). Furthermore, the activity does not seem to be sustained throughout the match, and intensity seems to be highest in the first ten minutes. In those studies, authors used the PlayerLoad[TM] metric. This metric is calculated with the sum of the accelerations across all axes of the internal tri-axial accelerometer during movement. However, this metric showed some inaccuracies in a recent study [9]. Indeed, this experiment aimed at verifying the validity of the PlayerLoad[TM] calculation against force platforms and at validating a new metric called Accel'Rate that would be more accurate. It was shown that although there is a good correlation between the PlayerLoad[TM] from the IMU and the calculation from the platforms, there is a significant average bias (from 17.1 to 226.0 a.u.). On the contrary, the Accel'Rate variable showed low biases (between -1 and 6.1 a.u.), while having a good correlation with the criterion value. In addition, the combined use of IMUs with LPS to determine body orientations is also a topic that has been studied in diverse activities [10, 11]. Determining these orientations would allow a quantification of forward, backward, and lateral movements and displacements. This could give a better idea of the efforts made during different phases of the game in handball and whether there are differences in the type of displacements by playing positions. It will also enhance load analysis and will help coaches to design more precise training exercise.

The current research therefore aimed at precisely quantifying external load in handball using LPS and IMUs. This technology allows to get individual data during competitive situation in a non-invasive method. The first objective was to develop and validate an automated phase and player specialty detection algorithm with reference to a regular video-based analysis. Secondly, this detection algorithm was applied on multiple games to automatically monitor physical demands per playing position and per game phases to investigate possible differences in load through variables like running distance or normalised distance but also through variables calculated from IMUs like Accel'Rate and time spent in each displacement direction.

## Material and methods

### Data sample

Data was collected on a Spanish first division handball club (highest level of competition in Spain) and covered 31 matches over two seasons (2019–2020 and 2020–2021). This club finished top 5 nationally over these two years. In this sample, all the games were played at home and we equipped all the players with a sensor including an IMU and linked to a LPS based in the club arena. Over those games, 27 different male handball players appeared in the team (average SD for height: 1.90 ± 0.06 m; weight: 91.9 ± 12.5 kg and age: 26.3 ± 3.9 years old). For each game, the group was composed of 14 to 16 players from the pool of 27. The players performed on average five handball training sessions and two to four strength and conditioning sessions per week during the season. Each session last from 1.5 to 2 hours and consisted in improving the physical profile in training (pre- or beginning of season) and/or performing technical or tactical situations (during the competitive season) that mimic game requirements. All players were identified as being at an elite or world class level [12].

### Data collection

Data come from the same sample used in Guignard et al [13]. We collected the data from the WIMU PRO system (RealTrack Systems S.L., Almería, Spain) with sensors placed on the back

of the athletes between the two scapulas at the level of the C7 vertebra. They were positioned in a special neoprene waistcoat to keep the sensor as fixed as possible during the game. Each sensor weighs 70 g and its dimensions are 81x45x16 mm (height/width/depth). This system has been subject to studies which ensure its validity and reliability [14–17]. Each sensor has a unique identifier and is affiliated with one player for an entire season to limit potential issues due to inter-unit reliability.

For the LPS data, the configuration used in this data collection is six antennas positioned around the court with the (0;0) coordinates matching a corner outside the court. As mentioned in the study by Bastida-Castillo et al. [14], this LPS UWB system works thanks to two subsystems: the reference system which is composed of the antennas acting as radio transmitters and receivers and the sensors carried by the participants. The system obtains the position of a sensor via triangulation of its position between the antennas and thus obtains its position in X and Y, alongside respectively the width and the length of the handball court (in m).

In parallel, each sensor contains an IMU, which includes a 3-axis accelerometer, 3-axis gyroscope, and 3-axis magnetometer. These sensors allow us to collect accelerations (in G), angular velocities (in deg/s) and magnetic field (in gauss) along X, Y and Z-axis. These axes are linked to the body of the player and therefore are different to the ones of the LPS. The X-axis points towards the head, the Y-axis points towards the player's right shoulder and the Z-axis points from the back towards the player's chest.

Ten minutes before the official starting time of each game, the WIMU were turned on for recordings. Ten minutes after the end of the game, all WIMU were turned off.

## Data transformation

LPS and IMU data were transformed from WIMU PRO software (SPRO, Real- Track Systems, 2018, version 964 [17]) format to a comma-separated value (.csv) format. The frequency of positional data (X, Y) ranged from 17 to 18 Hz and the frequency from IMU sensors were at 100 Hz. In the same way than Luteberget, Spencer, et al. [18], positional data was resampled from 17–18 Hz to 100 Hz using a cubic spline interpolation (i.e. of 3rd order) to have every variables on the same timeline and avoid synchronisation issues. With the same procedure than Guignard et al. [13], the positioning reference, initially defined outside the boundaries of the court, has been relocated to the centre of the court (on the median lines on both X and Y axes). by using the barycentre of the positions of the two GKs when they were in front of their goal. Thus, the extreme values of the court corresponded to −10 m and +10 m on the X-axis (court width) and −20 m and +20 m on the Y-axis (court length). The whole data treatment was performed on a custom-made Python algorithm (Python software version 3.8.8 through the Integrated Development Environment (IDE) Spyder v4.2.5).

## Game phase and player specialty recognition algorithm

**Data validation set.** Four handball games were randomly extracted from the sample and sequenced with Dartfish software (Fribourg, Switzerland) using a designed sequencing panel. Two of them were used to develop thresholds to detect game phase (see **Game Phase Detection**) and two of them were used for validation. A notational analysis [19] was performed to record game phases during the game, and then was compared to the LPS automatic data processing: start of the match, offensive phases, offensive transitions, stable defensive phases, defensive transitions, 7 m shots (penalty throws), presence of time-outs (either requested by the coaches or half-time break). Duration of offensive and defensive phases were computed and compared to those obtained from the automatic data treatment.

**Table 1. Presentation of variables used in the automatic algorithm.**

| Variables | Definition |
|---|---|
| N_Player | number of players on the court at each timeframe |
| Y_Pos_Court | average Y position of players on court at each timeframe |
| Y_Pos_Bench | average Y position of players on bench at each timeframe |
| Y_Speed | displacement speed of team players on court along Y-axis (length of court) at each timeframe |

**Data resampling and cleaning.** To reduce calculation time as all 16 players will be analysed simultaneously, the data were resampled from 100 Hz to 10 Hz for the phase recognition, merged with the positional data of all the other players and positioned on a single timeframe.

**Variables used.** Several variables were then derived to differentiate game phases (Table 1). To understand whether a player was on the court or not, a bench zone from 13.5m on both sides of the median line on Y and at 9.85m from the centre line on X was created.

**Game phase detection.** The thresholds for game phase detection were set empirically and step by step fine-tuned through testing them on two games of the sample (see **Data Validation Set**). For characteristics times and time outs, the number of players on the court was used (Fig 1). Some key moments of the game like the start of the game or the halftime break were initially detected in order to get a first level definition of the game (Table 2). When a team had more than nine players for a duration of 50 s to 90 s on the court during the first or the second half, a time out was detected.

To determine game phases (offensive play, defensive play, offensive and defensive transitions), three variables Y_Pos_Court, Y_Pos_Bench and Y_Speed were used. An example of

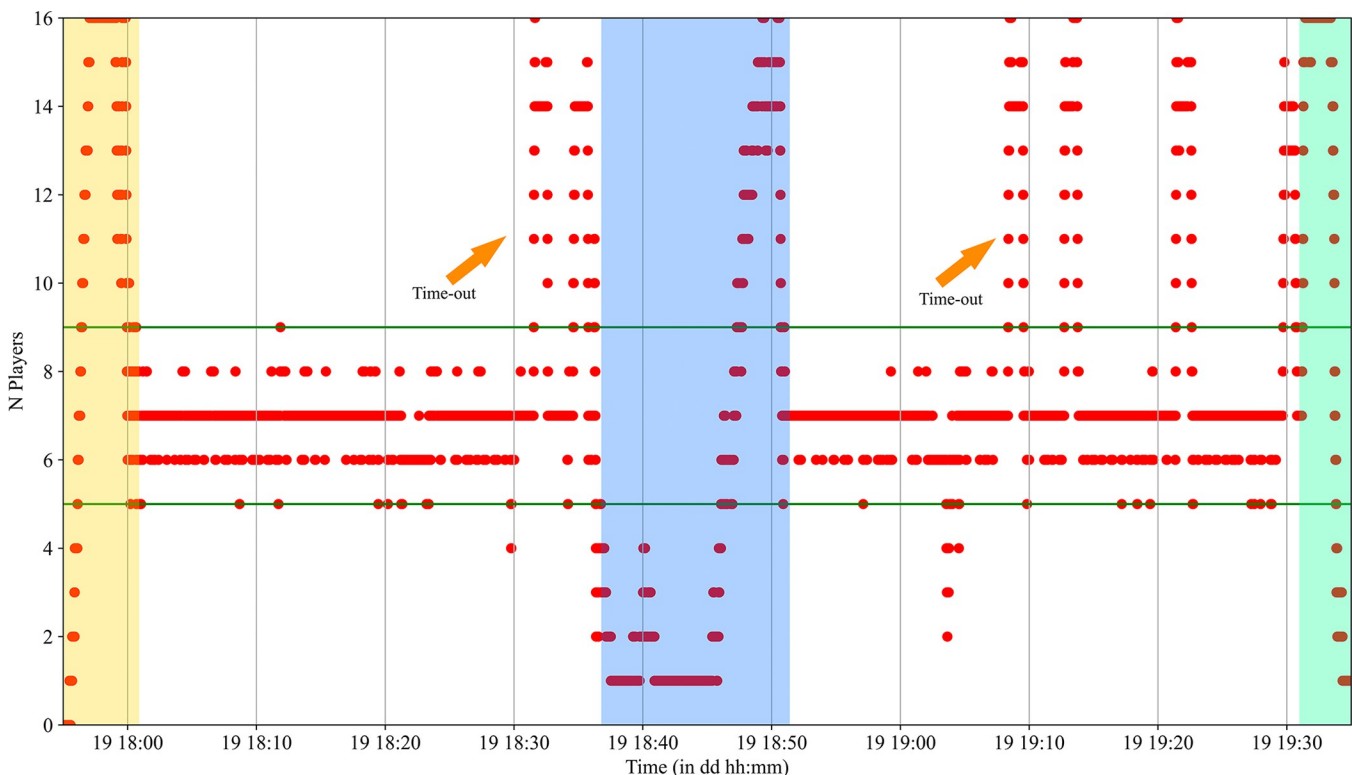

**Fig 1. Number of players on the court during a game.** The pregame is in yellow, the halftime break is in blue and the postgame is in green. Yellow arrows show time-outs periods.

**Table 2. Conditions to define starting and ending time of pregame, game or halftime.**

| Moment | Condition Time | Condition N_Players |
|---|---|---|
| Start Pre-Game | | N_Players = Max(N_Players) |
| End Pregame | t > Start Pre-Game | N_Players ≠ Max(N_Players) |
| Start Game | t > End Pregame | N_Players < = 7 |
| Start Half Time | t > Start Game | N_Players < = 3 |
| Start Half Time Warm-up | t > Start Half Time | N_Players > = Max(N_Players)-1 |
| Start Second Half | t > Start Half Time Warm-up | N_Players < = 7 |
| End Game | t > Start Second Half | N_Players < = 3 |

t, time of play; N_Players, number of players; Max, maximum.

Y_Pos_Bench and Y_Pos_Court are represented in Figs 2 and 3. First, in handball, a team bench is always positioned where the team is defending. If the team bench was positioned in negative Y values then an offensive play was considered if Y_Pos_Court was over +4.5m and a defensive play was considered if Y_Pos_Court was under -8m. On the other side, when the bench was in positive Y, an offensive phase was considered if Y_Pos_Court was under -4.5m and a defensive play was considered if Y_Pos_Court was over +8m. In between these phases, a transition phase should occur.

**Player specialty recognition.** To determine what was the specialty of the players (i.e., offensive play, defensive play, or both), a ratio between the time spent by each player in defensive phases, compared to the total time spent by the team in defensive phases was computed ($x = \frac{t_{defense\ PLAYER}}{t_{defense\ TEAM}}$).

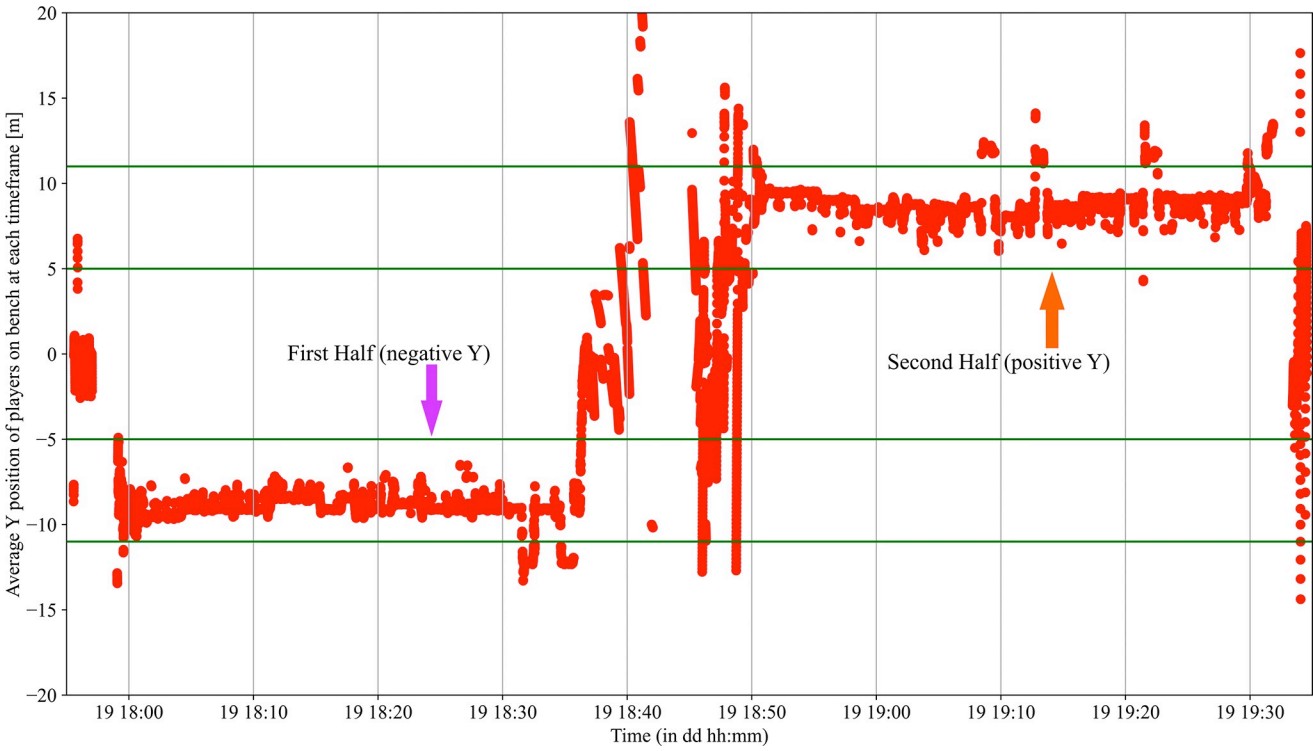

**Fig 2. Average Y position of players on the bench.** First half the bench was positioned in negative Y and second half in positive Y.

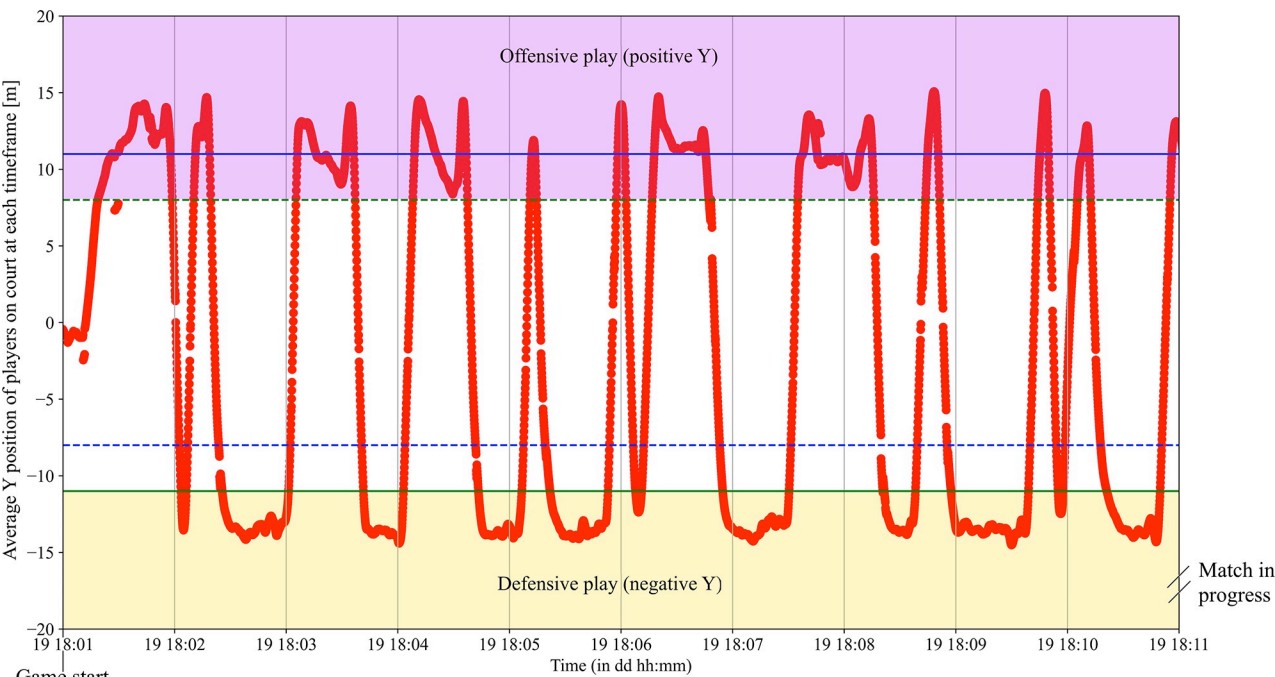

**Fig 3. An illustration of the average Y position of players on the court at the beginning of a match.** The defensive phases are shown in yellow, for negative average Y values, because the bench is positioned in negative Y (see Fig 2) and the offensive phases are in the purple zone (positive average Y values).

If a player was an offensive specialist, then this ratio is below 0.65.

For defensive specialist, first a fixed ratio was used. However, on some games the team spent a lot of time in defensive plays and therefore no defensive specialist were recognized (sensitivity issues). So, a variable threshold was designed (Table 3) based on the percentage of time spent by the team in defensive plays. A verification was done using video analysis to check the veracity of this calculation through the validation set.

## Computation of physical variables

**Displacement direction.** To characterise player's displacement direction (forward, backward, left, or right), Madgwick's orientation filter was applied on IMU data to get the shoulder orientation in a transverse plane and compare it to the orientation of the displacement of the player. First, to apply Madgwick's orientation filter at a later stage, the units of the sensors data must be changed as follows: accelerometer data in $m.s^{-2}$, gyroscope data in $rad.s^{-1}$ and magnetometer data in millitesla (mT). Secondly, to reduce noise in IMUs' signal, a low-pass Butterworth filter of $4^{th}$ order with a cut-off frequency of 10 Hz was applied [20]. Madgwick orientation [21, 22] filter from AHRS (Attitude and Heading Reference Systems) Python

**Table 3. Threshold for defensive specialisation.**

| % of effective time spent in defensive plays by the team | Ratio |
|---|---|
| <64% | 1.4 |
| 64% < p < 72% | 1.3 |
| >72% | 1.2 |

p, defensive plays.

toolbox was then applied to the raw inertial data. The results is a quaternion that was projected in the (X,Y) plane of the handball court to get the angle of the shoulder orientation from the player at each instant of the game. Finally, the difference between the orientation of displacement angle (obtained by calculating the arctan($\Delta$y/$\Delta$x) method through positional data) and the shoulder orientation was computed to distinguished between a forward, a backward or a lateral displacement. If the angle was between 45° and -45° it was considered a forward displacement, if the angle was between +45° and +135° it was considered as a displacement directed to the left, in between -45° and -135°, it was considered as a displacement directed to the right and finally between 135° and 180° or between -135° and -180°, it was considered as a backward displacement.

**Distance.**   From each player's positional data, the distances covered over time were calculated as the Euclidean distance from one coordinate (X;Y at n) to the subsequent one (X;Y at n —1). Those distances were computed for all the players during the whole match.

**Speed.**   A numerical differentiation over time was performed from the coordinates of each player on the court to obtain their instantaneous speed data. To detect the start and end of the stand, walk, jog, and sprint sections, the thresholds of Büchel et al. [23] were applied. Therefore, the following zones were considered as standing from 0 to 0.2 m.s$^{-1}$, walking from 0.2 to 2.0 m.s$^{-1}$, jogging from 2.0 to 4.0 m.s$^{-1}$, running from 4.0 to 5.5 m.s$^{-1}$ and finally sprinting when the instantaneous speed was over 5.5 m.s$^{-1}$.

**Accel'Rate.**   Accel'Rate was computed as a mechanical variable [9]. The formula is the following with $a_x$, the acceleration along the X axis, $a_y$ the acceleration along the Y axis and $a_z$, the acceleration of the Z axis (in m.s$^{-2}$) in the sensor frame of reference.

$$Accel'Rate = \frac{|\sqrt{(a_{x(t)})^2 + (a_{y(t)})^2 + (a_{z(t)})^2} - \sqrt{(a_{x(t-0.01)})^2 + (a_{y(t-0.01)})^2 + (a_{z(t-0.01)})^2}|}{100*g} \quad (1)$$

**Normalised variables.**   To avoid being biased by the time spent on the court by a player, all the distance covered and Accel'Rate were normalized at 10 minutes.

## Statistical analysis

Variables studied were the total distance covered in play (in m) per game normalised to 10 min (m x 10min), the total time in play (in min), the total Accel'Rate (in a.u.) per game and normalised to 10 min (a.u. x 10min), the percentage of total time in each displacement direction or in each speed zones (in %). Those values are presented as mean ± standard deviation. Phases where the player is on the bench or in time-out were removed from the study. To test for significant differences in playing positions or game phases, as the data were not meeting the conditions for homogeneity of variance (Levene's test), the Kruskal Wallis test was applied and Dunn's post-hoc test was implemented when main effects were significantly lower than the p<0.05 threshold. For all statistical tests JASP software Version 0.16.0 was used.

## Results

A total of 31 handball matches were included in the present study. It corresponds to the analysis of 491 data sets of players (positional and inertial data). If player's playing time was less than 4 minutes, this was considered not representative enough of a classic handball activity. 28 players data were therefore removed from the study bringing the total to 463 raw data.

The following distribution on player positions was observed: 120 wing players (Wings), 40 goalkeepers (Goalkeeper), 73 versatile line (Pivot_Both), 160 versatile backs (Backs_Both), 37

defensive backs or defensive line player (Pos3_Def), 33 offensive backs (Backs_Off). There is no offensive line players in this study. As defensive backs and defensive line players were covering the same position on the court (position 3 in defensive play), they were put in the same category.

## Phase recognition algorithm

The validation was performed by comparing the phase sequences from the same game between the automated phase recognition algorithm and the notational analysis. On average, $98.95 \pm 0.88\%$ of the game time was correctly recognised (Table 4). Two main types of error were found: wrong detection (i.e., when the phase of play was not the same type as the one on the video), and non-detection (i.e., when the phase on the video was simply not detected by the automated recognition).

## Physical demands

The average $\pm$ SD playing time by a player in this team was $30.4 \pm 12.6$ minutes, the average distance covered was $2388 \pm 871$ m, the average pace was $821.4 \pm 170$ m/10min and the normalised Accel'Rate was $82.7 \pm 22.9$ u.a./10min. There was a prevalence of time spent walking with $71.3 \pm 5.2\%$ and time spent in the forward displacement zone with $40.2 \pm 6.1\%$.

**Per position.** Descriptive statistics per playing position are given in S1 Table, whereas Dunn's post-hoc test for Distance and Accel'Rate variables are given in S2 and S3 Tables, respectively. Goalkeepers were the players spending the most of their playing time standing, $19.2 \pm 3.2\%$ compared to $7.0 \pm 2.8\%$ for the other players. An average offensive back player had the highest percentage of playing time spent in jogging ($23.7 \pm 4.0\%$ vs. $13.9 \pm 2.4\%$ for the rest of the team excluding goalkeepers) and running ($8.2 \pm 1.6\%$ vs. $5.6 \pm 1.8\%$ for the rest of the team excluding goalkeepers). The percentage of time spent by wing players in sprint was the highest with $3.5 \pm 0.9\%$ of the time played compared to $1.1 \pm 0.6\%$ for the rest of the team excluding goalkeepers. For the type of displacement, there was no significant difference in this position classification (Fig 4).

The Kruskal Wallis test showed the significant effect of the position factor for the variables normalised distance ($H(5) = 310.265$, $p < .001$) and normalised Accel'Rate ($H(5) = 228.032$, $p < .001$) (Figs 5 and 6).

**Per game phase.** In this part, goalkeepers' data were not taken into account. Descriptive statistics per game phases are given in S4 Table, whereas Dunn's post-hoc test for Distance and Accel'Rate variables are given in S5 and S6 Tables, respectively. An average player from this team spent most of his time in defensive phases ($16.7 \pm 6.9$ minutes) followed by offensive phases ($6.9 \pm 3.9$ minutes). The time spent in offensive ($3.5 \pm 1.5$ minutes) and defensive transitions ($2.4 \pm 1.1$ minutes) was similar and remained relatively low. The rest of the time, players are on the bench. Players spent very little time standing ($2.1 \pm 2.5\%$) or walking ($39.6 \pm 11.6\%$) in transitions. Conversely, during defensive phases, players spend 90% of the time standing ($8.5 \pm 2.5\%$) or walking ($83.1 \pm 2.9\%$). There was more forward displacement in transition

**Table 4. Quality of the data validation set recognition for handball phases.**

| Match identifier | 1 | 2 | 3 | 4 |
|---|---|---|---|---|
| Wrong detection time (s) | 41.4 | 89.3 | 33.1 | 2.9 |
| Non-detection time (s) | 10.2 | 17.6 | 0 | 0 |
| Total coded time (s) | 4509.9 | 4903.7 | 4145.7 | 3908.6 |
| % of time with correct recognition | 98.86% | 97.82% | 99.20% | 99.93% |

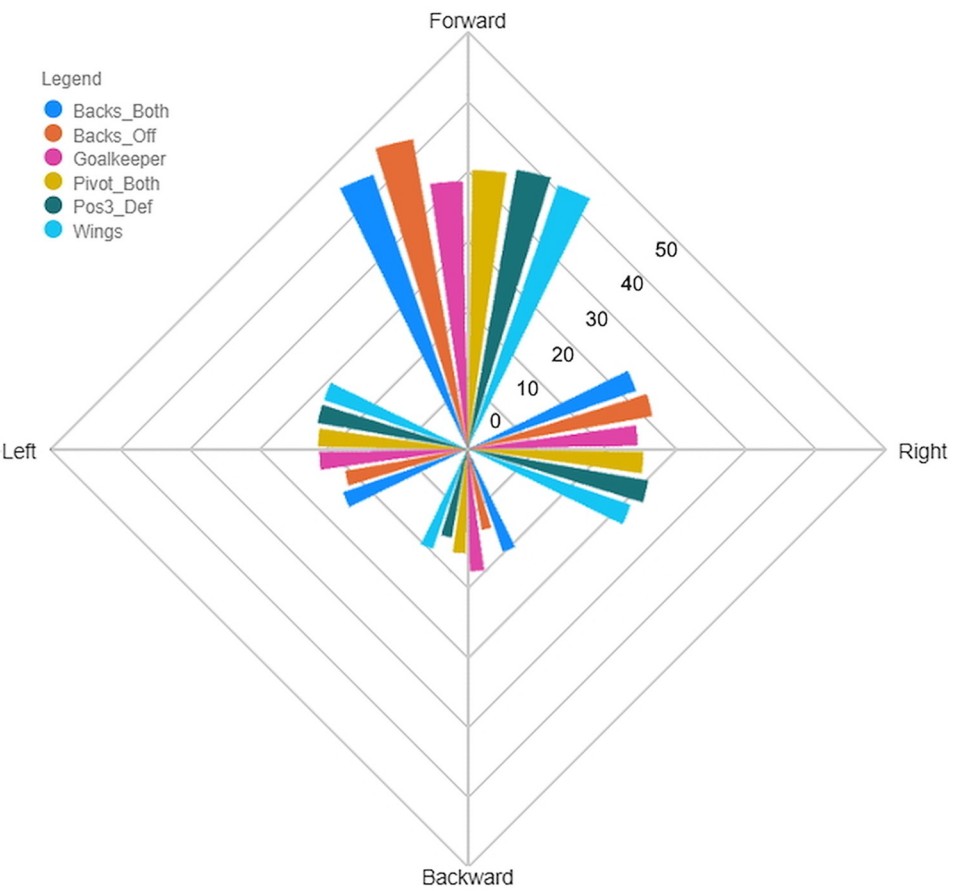

**Fig 4. Percentage of time spent in each displacement direction by position.**

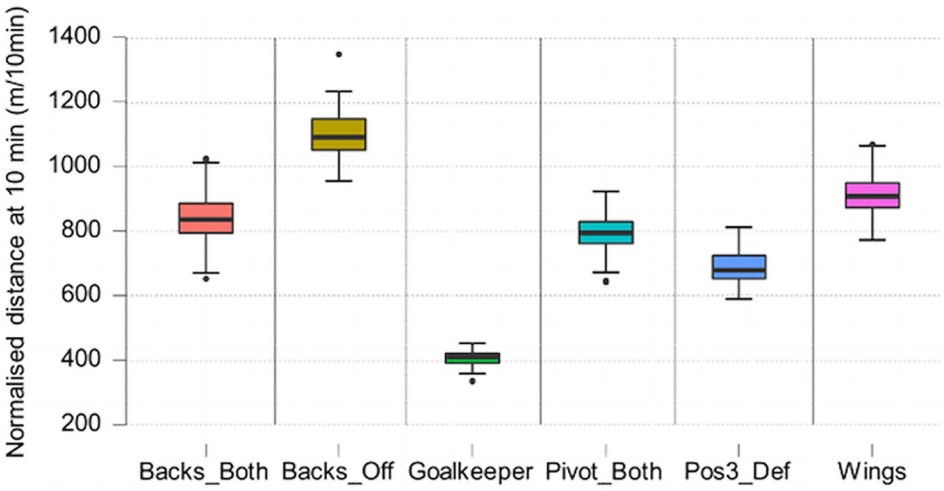

**Fig 5. Boxplot of normalised distance per position.** Highly significant differences between positions (p < .001) except Goalkeeper and Pos3_Def where it is significant at p <0.05.

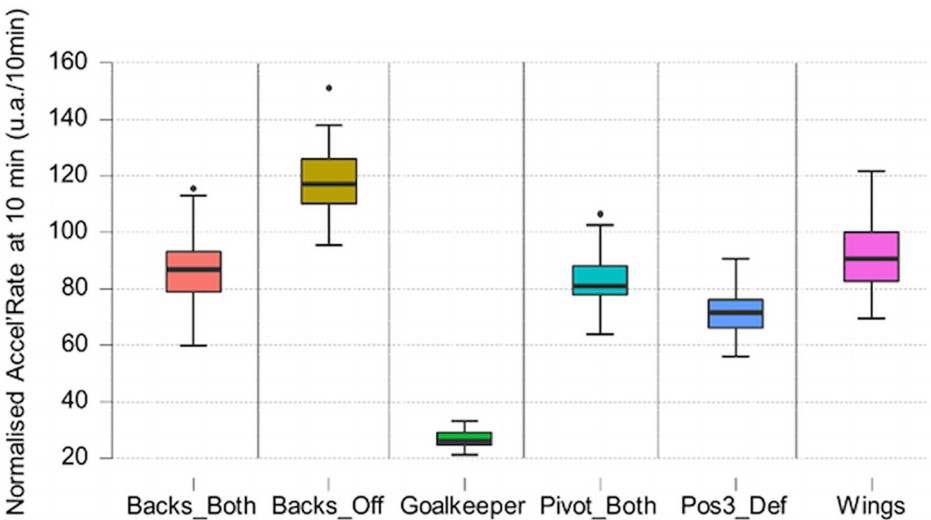

**Fig 6. Boxplot of normalised Accel'Rate per position.** Strong significant differences between positions (p < .01) except and Backs_Both—Pivot_Both where it is significant at p <0.05.

phases (49.1 ± 13.3%) compared to stabilized phases (38.6 ± 7.1%) (Fig 7). The comparison between offensive and defensive transitions showed more backwards displacements during defensive transitions as compared to offensive transitions (16.8 ± 9.2% during defensive transition versus 8.3 ± 5.0% during offensive transition).

The Kruskal Wallis test showed a significant effect of the game phase factor for the variables normalised distance per 10 minutes (H (3) = 1349.839, p < .001) and normalised Accel'Rate per 10 minutes (H (3) = 1161.856, p < .001) (Figs 8 and 9).

**Per game phase and positions.** The time spent in each phase and normalised distance in each phase per playing positions are shown in Figs 10 and 11. Goalkeepers had a normalised distance of 577 ± 84 m/10min in transitions and 358 ± 44 m/10min on stabilized phases. They also have the highest playing time in each phase (4.3 ± 1.5 minutes in defensive transitions, 6.1 ± 2.1 minutes in offensive transitions, 28.3 ± 9.5 minutes in defensive plays and 10.1 ± 5.7 minutes in offensive plays). Wing players had the highest values on normalised distance during transitions respectively, 1940 ± 234 m/10min on defensive transition and 1886 ± 234 m/10min on offensive transition compared to the rest of the team (excluding Goalkeepers) that has an average of 1471 ± 243 m/10min on defensive transitions and 1583 ± 186 m/10min on offensive

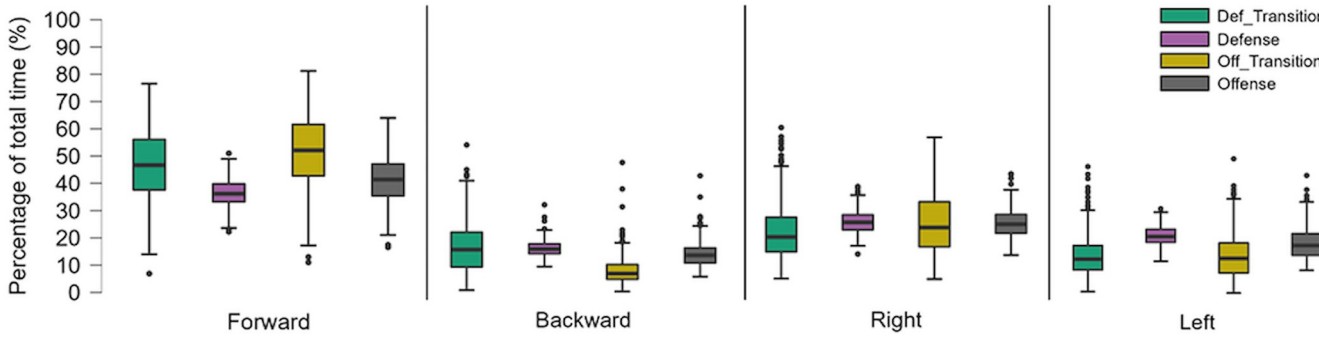

**Fig 7. Mean ± SD of time spent per displacements direction (in %) for each game phase.**

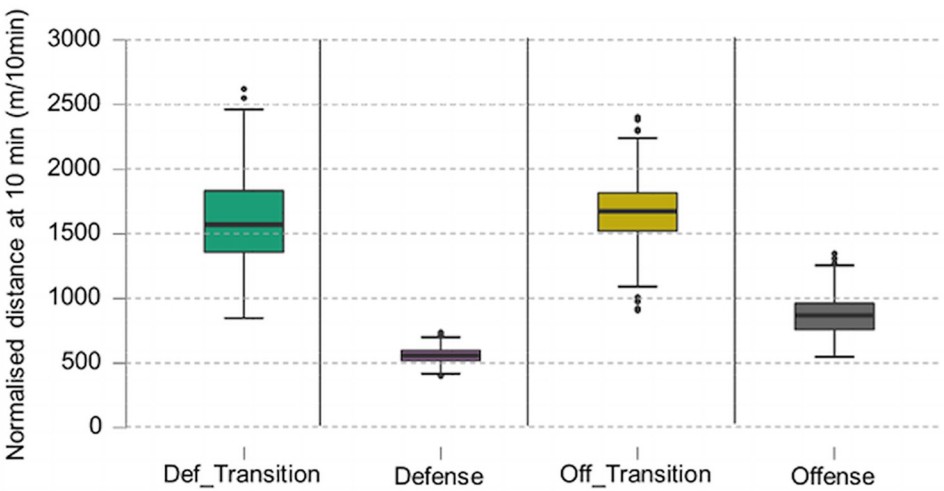

**Fig 8. Boxplot of normalised distance per game phase.** Highly significant differences between groups (p < .001) except Def_transition and Off_transition where it is significant at p <0.05.

transitions. On stabilized offensive plays, offensive backs had the highest normalised distance with 1046 ± 108 m/10min and the highest playing time behind goalkeepers, 7.9 ± 4.4 minutes. On the opposite, without considering goalkeepers, line and wing players had the lowest normalised distance with respectively 741 ± 78 m/10min and 794 ± 111 m/10min. The normalised distance per players and phases showed no significant differences during defensive phases (i.e., with an average of 559 m ± 54 m/10min). The standard deviation for normalised distance on Position 3 defensive players in offensive plays is big because we have only 3 sets of data with this combination in the database.

## Discussion

The aim of this study was to go further than an analysis of match demands through local positioning system or inertial measurements units. This technology is the source of efficient and

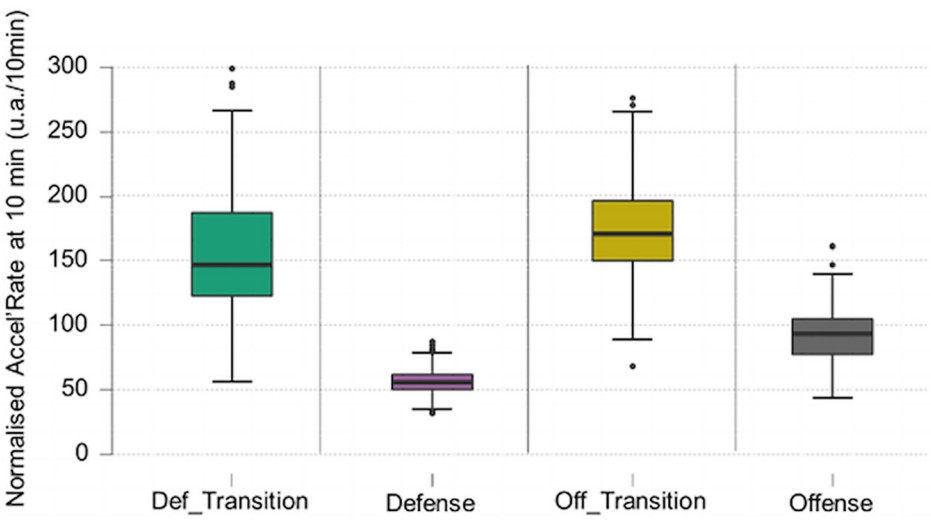

**Fig 9. Boxplot of normalised Accel'Rate per game phase.** Highly significant differences between groups (p < .001).

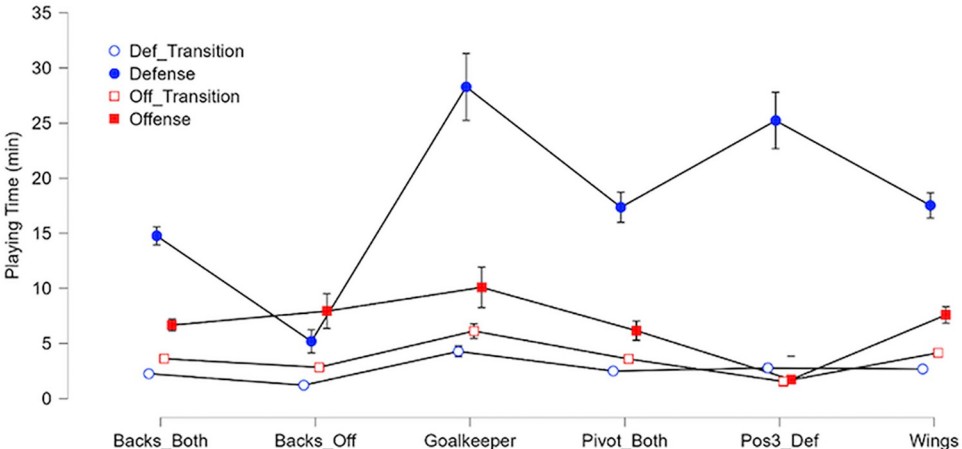

**Fig 10. Plots of time played per position/phase of play with 95% confidence interval.**

appropriate data collection with the demands of modern handball. By using them, we aimed at developing an automatic tool to classify game phases, check its classification quality, and quantify the match demands per game phases automatically. We also wanted to create distinction between players through positions (goalkeepers, wing, back or line players) and specialty (offensive, defensive or none).

First, an automated phase recognition algorithm was designed to detect game phases through positional data in handball. In the same idea, automation of defensive patterns was already developed by Guignard et al [13]. When the phase recognition algorithm was applied to the validation sets, there is an average of 98.95% of the time where the phase was correctly recognized. In view of these results, it was considered that it was reliable and could automate the recognition of game phases during handball games. This type of algorithm can automate tactical and physical analysis and avoid time-consuming video analysis.

The second purpose of this article was to look at physical data in match conditions through classic and innovative variables to determine if there are some differences. Regarding game

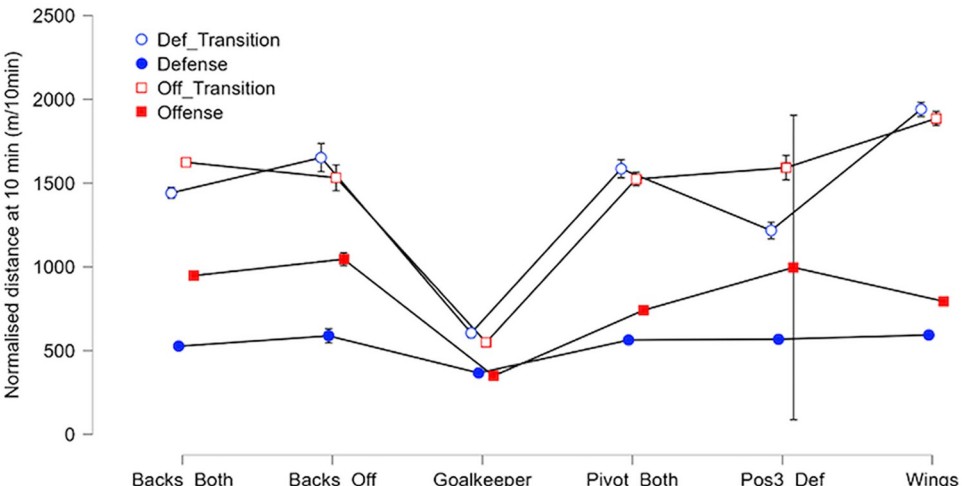

**Fig 11. Plots of normalised distance per position/phase of play with 95% confidence interval.**

phases, the classical division used in former studies were 'offensive play' or 'defensive play' [5, 6]. In contrast to these studies, transitions were considered, whether they were offensive or defensive. They represented 20% of the average playing time (respectively 11.9% and 8.1%). These phases are important to be distinguished because there are larger values for normalised distance and normalised Accel'Rate in transitions phases (respectively 1671.26 ± 242.02 m/10min and 174.53 ± 35.22 u.a./10min for offensive transitions and 1604.51 ± 319.97 m/10min and 154.34 ± 45.22 u.a./10min for defensive transitions) compared to offensive phases (870.34 ± 145.68 m/10min and 92.45 ± 20.67 u.a./10min) and defensive phases (558.82 ± 53.88 m/10 min and 56.47 ± 9.00 u.a./10min). On displacement directions, players tend to go more forward. There is a difference between stabilized phases and transition phases with more vertical displacement (forward or backward) in transition compared to stabilized phases. Also, one of the main attributes to differentiate offensive and defensive transition is their backward and forward displacement percentage with more backward displacement percentage in defensive transitions. To the best of our knowledge, this type of analysis with a separation between transition and stabilized phases has never been conducted in handball.

In the literature, studies have shown distinctions in physical demands per playing positions (goalkeepers, wing players, back players and line players) in the past ten years [3–6, 23, 24]. The position of a player does not seem to have an impact on the percentage of time spent in each type of displacement. Significant differences were detected on two variables: normalised distance and normalised Accel'Rate. This effect is present in between all the positions. It is consistent with most studies since 2010.

Goalkeepers have a different role than all other players on the field. They have the highest playing time (49.43 ± 16.7 minutes) similar to what was published by Cardinale et al [25]. They cover an average distance of 2km per game like the position 3 defensive players, but the physical demands are less intense. Normalised distance is 405.6 ± 28.4 m/10min which is two times lower than the average for the rest of players and normalised Accel'Rate is 26.9 ± 3.3 a. u./10min which is three times lower than the average for all the players.

Considering field players, wingers have an average playing time about 30 minutes, cover the greatest total distance (2926 ± 998.79 m) and have the highest average Accel'Rate per game (297 ± 102.21 a.u.). Physical demands on this position are higher than on the other positions in transition phases, which explains the high percentage of time spent in sprint. To compensate for this, they have a lower demand on stabilized phases (Fig 10). Versatile backs and line players have similar profiles compared to other roles. Average distance and total load on these positions are close (about 2350 m and 240 a.u.). Such results should of course be linked to the playing style: for the current study, the video data validation set revealed that the team has the skills to very early recover the balls from the opponent, with a fast forward move in counterattacks. This style may influence the present results regarding the distance covered by players especially during transition phases (fast counterattacks to create danger on the opponent's goal and withdraw quickly to ensure a new defensive play).

When comparing versatile and offensive back players, there is a significant difference in several variables. Offensive backs have the lowest playing time with an average of 15.06 minutes compared to 27.64 minutes for versatile back players. Moreover, the normalised distance and the normalised Accel'Rate is more important in offensive plays than in defensive plays (+56% on normalised distance and +64% on normalised Accel'Rate). This explains why offensive back players have higher values on these two variables than versatile back players in the study. The game therefore requires them to cover less total distance but to have a higher volume in the little playing time they have.

There are also some differences when comparing defensive position 3 players to versatile back and line players. Defensive specialists have an average playing time of 30.11 minutes like

versatile back (27.64 min) and line players (29.97 min). As shown previously, defensive play is the phase of the game with the lowest physical demand in this study. Apart from goalkeepers, defensive specialists are therefore the players with the lowest values on the normalised and total distance and Accel'Rate. A defensive specialist cover less total distance (2062 m) and have less total Accel'Rate (214 u.a.) than versatile back and line players. However, it is a position where physical impacts (pressure with the opposing line players or contact with launched back players) can be important [25] and which may soon be analysed with the development of embedded technologies. For instance, a recent study combined the raw recordings of an embedded GPS and tri-axial accelerometer in professional rugby union to automatically detect collisions from a custom algorithm sensible to a minimum threshold of 5 $g$ [26]. It would be interesting to look at other variables like the quantity and intensity of jumps (see for instance Póvoas et al. [4]), the number and intensity of accelerations or decelerations (as previously done in Font et al. [27]) or estimating the internal load to better quantify the total load associated with these positions. Indeed, the monitoring of the mean and maximal Heart Rate (HR) in official handball matches in a previous study [4] revealed that back players and line players had higher mean (respectively 84 ± 9% and 83 ± 9% of HRmax) and higher peak (respectively 96 ± 4% and 98 ± 2% of HRmax) values compared to other playing positions in the team.

From the present results, not separating offensive and defensive specialists from versatile backs and line players in previous studies may not be accurate and may not fit with modern handball which sees more and more specialisation in player profiles. To the best of our knowledge, this type of analysis with a clear distinction between defensive, offensive, and versatile players at the back and line position has been done in part by Manchado and al. (2020) [6] but only with defensive players and Luteberget and Spencer (2017) [7] only on intensity events. These kinds of profiles can help the design of training programs to better suit the individual demands of the position.

This study must be interpreted in light of some caveats. On a tactical side, the present results are dependent on the playing style of the team and for that purpose it is pertinent to monitor accurately the players' orientation, position and game phases to analyse representative physical demands of the game. As all our data are coming from the same team, it should be noted that some variables could have influence the results like all our games analysed were at home [28] or the level of competition, even if we tried to reduce this by analysing as much games as possible. Also in traditional handball, each position of players (e.g. back, line or wing players) retained their role either in offensive plays and defensive plays, but with the evolution of handball, player profiles might have to be split with an offensive and a defensive position. For instance, it is now frequent to see back players that can be positioned on a wing position in defensive plays. On a physical side, new innovative variables were used to quantify accurately the game's physical demands like the shoulder orientation, but it would be ideal to validate this calculation to quantify the error between our calculation and the real shoulder orientation. Finally, other system can give positional data on the Z-Axis that would take into account the impacts of repetitive jumps on physical demands.

## Conclusion

The purpose of this study was to develop an algorithm to recognize on court-activity and player specialty to give a context to the quantification of physical demand during handball competitions. It was also to use jointly LPS and IMUs to combine motion and mechanical data all at once. The underlying idea was to use this new technology to bring a new insight on match demands. Results indicate that our game-phase recognition was accurate and could gave information on the tactical organisation of the team studied to give context on the

physical demands. It also underlines that game phase has a significant impact on physical demands and that transition phases must be separated to stabilized phases in future analysis. Finally, our results showed that not only physical demands are different in between playing positions, but it can also be different between specialists (defensive, offensive, both end of the floor) of a same position. In the future, with the development of machine learning video analysis, data from both teams and the ball in terms of position could become available which can give more insights to understand physical match demands regarding tactical behaviour (defensive schemes, offensive system), game events (numerical superiority or inferiority), game context (team is winning or losing). Crucial tactical information such as the number of substitutions, the duration of the different offensive and defensive phases of the game would be available continuously. More broadly, it could also automate the detection of shots, passes and correlate that to the player sensors (IMU). All-together, this information would help monitoring modern handball to individualize the cost of each of these actions/sequences.

As practical applications, the present information could help the decision-making of the head coach and strength and condition coach to quantify the physical demands for each player in the game, and hence to determine with accuracy the recovery time the players can benefit from. This is an essential step in the monitoring of the players' efforts, in order to refine the load management that teams have to do throughout the season to minimize risk injury and increase performance on key matches. It also can help to anticipate the rotations that teams must do in-game with the unlimited substitutions opportunities in handball.

## Supporting information

**S1 Table. Descriptive statistics per playing position.**
(DOCX)

**S2 Table. Dunn's post-hoc test for the variable distance normalised for positions factor.**
(DOCX)

**S3 Table. Dunn's post-hoc test for the variable Accel'Rate normalised for positions factor.**
(DOCX)

**S4 Table. Descriptive statistics per game phases.**
(DOCX)

**S5 Table. Dunn's post-hoc test for the variable distance normalised for game phases factor.**
(DOCX)

**S6 Table. Dunn's post-hoc test for the variable Accel'Rate normalised for game phases factor.**
(DOCX)

## Author Contributions

**Conceptualization:** Brice Guignard, John Komar.

**Formal analysis:** Thomas Lefèvre.

**Funding acquisition:** Brice Guignard.

**Methodology:** Thomas Lefèvre, Brice Guignard, John Komar.

**Project administration:** Brice Guignard, John Komar.

**Resources:** Xavier Reche, Roger Font, John Komar.

**Software:** Thomas Lefèvre.

**Supervision:** Brice Guignard, Claude Karcher, John Komar.

**Validation:** Thomas Lefèvre.

**Visualization:** Thomas Lefèvre.

**Writing – original draft:** Thomas Lefèvre.

**Writing – review & editing:** Thomas Lefèvre, Brice Guignard, Claude Karcher, Xavier Reche, Roger Font, John Komar.

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
