## [Decision Letter · Decision Letter 0]

15 May 2023

PONE-D-23-10764A deep dive into the use of Local Positioning System in professional handball: Automatic detection of players’ orientation, position and game phases to analyse specific physical demandsPLOS ONE

Dear Dr. Guignard,

Thank you for submitting your manuscript to PLOS ONE. After careful consideration, we feel that it has merit but does not fully meet PLOS ONE’s publication criteria as it currently stands. Therefore, we invite you to submit a revised version of the manuscript that addresses the points raised during the review process.

We look forward to receiving your revised manuscript.

Kind regards,

Rabiu Muazu Musa, PhD

Academic Editor

PLOS ONE

Journal Requirements:

2. Please expand the acronym “CETAPS” (as indicated in your financial disclosure) so that it states the name of your funders in full.

Reviewers' comments:

Reviewer's Responses to Questions

**Comments to the Author**

1. Is the manuscript technically sound, and do the data support the conclusions?

Reviewer #1: Yes

Reviewer #2: Yes

2. Has the statistical analysis been performed appropriately and rigorously? 

Reviewer #1: Yes

Reviewer #2: I Don't Know

3. Have the authors made all data underlying the findings in their manuscript fully available?

Reviewer #1: Yes

Reviewer #2: Yes

4. Is the manuscript presented in an intelligible fashion and written in standard English?

Reviewer #1: Yes

Reviewer #2: Yes

5. Review Comments to the Author

Reviewer #1: Dear Authors.

The following is a review of the article entitled “A deep dive into the use of Local Positioning System in professional handball: Automatic detection of players’ orientation, position and game phases to analyse specific physical demands”. Thank you very much for thinking of me as a reviewer for this study.

After carefully reading the manuscript, I set forth comments and suggestions for the authors:

Your work is very interesting, and I congratulate the authors by a unique study. I have

some suggestion to improve clarity of your work that highlight below. Please check:

Abstract: Correct.

Line 46. The aim of your study is to analyse the quantification of the external load during an elite men's handball match. It is recommended that data on the quantification of the external load on the variables analysed appear in the summary.

Keywords: It is recommended not to repeat words from the title in the keywords.

Introduction: The introduction contains sufficient information to understand the topic and to justify the study.

Materials and Methods:

Lines 96-97: I guess that Spanish first division handball club is the highest handball level in the country, but it should be specified. Moreover, a more definition of the athlete’s competitive level could be provided following the indication of this work: McKay AKA, Stellingwerff T, Smith ES, Martin DT, Mujika I, Goosey-Tolfrey VL, Sheppard J, Burke LM. Defining Training and Performance Caliber: A Participant Classification Framework. Int J Sports Physiol Perform. 2022 Feb 1;17(2):317-331. doi: 10.1123/ijspp.2021-0451.

Line 215. Why is the average Y position of the players on court from 5 minutes before the start of the match to 10 minutes after the start of the match? Shouldn't only the match be analysed?

Results:

-Line 234. If the core body temperature recorded before and after all heating protocols is shown first in the results, should this be one of the objectives of the study?

-Line 210. Clarify Table 1. Effects of different PAP warm-ups on RSA (Mean±SD). Positive effect size for a negative percentage difference?

Figures. Must show the unit of measurement.

Discussion:

Line 483: Correctly justify this statement. “Before backs players were place in the center of the court in offensive plays and defensive plays but with the emergence of short center backs, back players can be positioned on a wing position in defensive plays to compensate their lack of height. So, to follow the evolution of handball, player profiles might have to be split with an offensive and a defensive position”. Is it a limitation of the study?

Reorder the discussion in order of appearance of the results. To what may be due to each of the results obtained?

Please, provide practical applications.

Conclusions:

Very ambiguous conclusions. They should not be compared with other authors, this should be done in the discussion.

In your conclusions you should specify What conclusions do you draw from the study? Do the results give a correct answer to the objectives of the study? What conclusions do your results contribute to science?

References: Correct.

Thank you so much for your consideration.

Reviewer #2: I think it's good research.

I would only add some more data to "Data sample" in relation to training, such as: number of weekly training sessions, duration of training, time dedicated to improving the physical profile in training, time dedicated to improving technical situations -tactics, specify if the players have their nutrition controlled by a nutritionist, hydration level in training, etc.

For future studies, it would be great to be able to relate the study's conclusions to contextual factors, such as: what happens when the team plays at home or away, what happens when the team loses or wins, what happens when the team is playing with one more player or less player, etc.

6. PLOS authors have the option to publish the peer review history of their article (what does this mean?). If published, this will include your full peer review and any attached files.

Reviewer #1: **Yes: **Demetrio Lozano Jarque

Reviewer #2: No

<quillbot-extension-portal></quillbot-extension-portal>

---

## [Author Response · Author response to Decision Letter 0]

5 Jul 2023

PONE-D-23-10764

A deep dive into the use of Local Positioning System in professional handball: Automatic detection of players’ orientation, position and game phases to analyse specific physical demands

Response of the authors will be marked in red in this document.

Journal Requirements:

1. Please ensure that your manuscript meets PLOS ONE's style requirements.

Response: The revised manuscript and attached documents were created as a function of PLOS ONE’s style requirements.

2. Please expand the acronym “CETAPS” (as indicated in your financial disclosure) so that it states the name of your funders in full.

Response: It was done according to the Journal Requirements and specified in the revised cover letter.

Response: The data presented in the current paper are owned by a third-party organization. For that reason, the data will be only available upon request to Dr. John Komar (National Institute of Education, Nanyang Technological University), or Dr. Brice Guignard (Faculty of Sport Sciences, University of Rouen Normandie), with an explanation of the purpose of the request.This is now specified in the revised cover letter.

Response: The captions were included as requested. The Supporting Information guidelines were scrupulously followed for the re-submission.

Reviewer #1: Dear Authors.

The following is a review of the article entitled “A deep dive into the use of Local Positioning System in professional handball: Automatic detection of players’ orientation, position and game phases to analyse specific physical demands”. Thank you very much for thinking of me as a reviewer for this study.

After carefully reading the manuscript, I set forth comments and suggestions for the authors: Your work is very interesting, and I congratulate the authors by a unique study. I have some suggestion to improve clarity of your work that highlight below.

Response: The authors appreciate the comments and review. We seek to respond appropriately and correct the text when it was necessary.

Please check:

Abstract: Correct.

Line 46. The aim of your study is to analyse the quantification of the external load during an elite men's handball match. It is recommended that data on the quantification of the external load on the variables analysed appear in the summary.

Response: We added more numbered information on our results in the summary.

Keywords: It is recommended not to repeat words from the title in the keywords.

Response: New keywords were added in the revised version of the manuscript, as follows: Keywords: IMU, position, external workload, performance analyses, algorithm.

Introduction: The introduction contains sufficient information to understand the topic and to justify the study.

Response: The authors appreciate the comment.

Materials and Methods:

Lines 96-97: I guess that Spanish first division handball club is the highest handball level in the country, but it should be specified. Moreover, a more definition of the athlete’s competitive level could be provided following the indication of this work: McKay AKA, Stellingwerff T, Smith ES, Martin DT, Mujika I, Goosey-Tolfrey VL, Sheppard J, Burke LM. Defining Training and Performance Caliber: A Participant Classification Framework. Int J Sports Physiol Perform. 2022 Feb 1;17(2):317-331. doi: 10.1123/ijspp.2021-0451.

Response: This information has been added to the article and thank you for the reference.

Line 215. Why is the average Y position of the players on court from 5 minutes before the start of the match to 10 minutes after the start of the match? Shouldn't only the match be analysed?

Response: In the physical analysis, we do not use the data before the start of the match. Data of players on court before the game was used to automate the detection of the start of the match. To avoid confusion, we have shortened the graphic on Figure 3 to show only in-game behaviors and correct our graph description. Figure 3 is only a close zoom; thus, the clock of the match is still in progress after the timeline indicated in this figure.

The average Y position of the players was therefore computed at each timeframe of the match, as specified in Table 1.

Results:

-Line 234. If the core body temperature recorded before and after all heating protocols is shown first in the results, should this be one of the objectives of the study?

Response: We are sorry, but it seems that this comment is not related to our article.

-Line 210. Clarify Table 1. Effects of different PAP warm-ups on RSA (Mean±SD). Positive effect size for a negative percentage difference?

Response: We are sorry, but it seems that this comment is not related to our article.

Figures. Must show the unit of measurement.

Response: All figures were carefully reviewed according to this comment. Figure 3 was modified to increase clarity of information.

Discussion:

Line 483: Correctly justify this statement. “Before backs players were place in the center of the court in offensive plays and defensive plays but with the emergence of short center backs, back players can be positioned on a wing position in defensive plays to compensate their lack of height. So, to follow the evolution of handball, player profiles might have to be split with an offensive and a defensive position”. Is it a limitation of the study?

Response: 

This is not a limitation of the study, but rather an observation that modern, high-level handball is constantly changing. This can be linked to rule changes such as the one in July 2016: goalkeepers can give way to an extra player when attacking. This rule makes the game faster, more focused on getting the ball back quickly for the defending team, in order to immediately bring the danger to the opponent's goal. This means that players need to be fast in order to make many transitions during a match. Classic positions such as back players, line players, wing players etc. are no longer as entrenched as they were in the past from one side of the court to the other, and for this reason hybrid profiles may emerge. For this reason, we highlighted that several players with the same position on the court have not necessarily the same physical requirements during the game since they performed different task (only offense, only defense, both end of the court). Thus, it is necessary to measure the precise physical demands of each player according to his profile and not only his position on the field to reveal his maximum potential in competition. For these reasons, it might be interesting to consider several physical preparation processes for such players.

To avoid confusions, the section was rephrased.

Reorder the discussion in order of appearance of the results. To what may be due to each of the results obtained?

Response: After considering the comment of the reviewer, we finally prefer not to change the order of our paragraphs in our discussion appear. This order has been chosen because we believe that it makes the most sense from a scientific point of view. When analyzing the results of the game phases we noticed that the physical demands during the defensive play are lower on the normalized distance in contrast to the other game phases. Moreover, the demands on this phase do not seem to be position dependent. For us, this kind of difference in physical demands leads to the differentiation of physical demands between specialists of the same position. Therefore, we saw important differences in physical demands between a defensive back, an offensive back or a back playing on both sides of the field. Our hypothesis is if an offensive back has a higher normalized distance than a versatile back, who in turn has a higher normalized distance than a defensive position 3 player, it should come from their % of total playing time in each phase which are totally different from the others as we can deduct from Fig 10. We therefore wanted to comment first on the results from the game phases before discussing those on the positions and on specialists in order to have a logical order on the analysis.

Please, provide practical applications.

Response: Practical applications are now clearly written at the end of the conclusion section. More, we added the following statement: quantifying the physical demands for each player in the game is an essential step in the monitoring of their efforts, which allow to determine with accuracy the recovery time they can benefit from.

Conclusions:

Very ambiguous conclusions. They should not be compared with other authors, this should be done in the discussion.

Response: The reference to the work of Povoas was deleted from the conclusion and added in the discussion, as requested. We also put limitations in the discussion to keep the essential part for the conclusion. 

In your conclusions you should specify 

What conclusions do you draw from the study? 

Do the results give a correct answer to the objectives of the study? 

What conclusions do your results contribute to science?

Response : We rewrite most part of the conclusion to answer your questions. Thank you.

References: Correct.

Thank you so much for your consideration.

Response: Thank you for reviewing the paper.

Reviewer #2: I think it's good research.

Response: Thank you for this positive comment.

I would only add some more data to "Data sample" in relation to training, such as: number of weekly training sessions, duration of training, time dedicated to improving the physical profile in training, time dedicated to improving technical situations -tactics, specify if the players have their nutrition controlled by a nutritionist, hydration level in training, etc.

Response: Thank you for the comment. Some of this crucial information was added in the “Data sample” section of the revised manuscript.

Regarding the time dedicated to improving the physical profile in training, this was highly dependent on the moment of the season: this is mainly the objective of the pre-season (e.g. 2 months before the start of the competitions) or the very start of the season. Later on, the training sessions are no more dedicated to improving the physical profile, rather, the objective is mainly to maintain the physical form. Regarding the time dedicated to improving technical or tactical situations, this is once again highly dependent on the moment in the season. Even if we may consider a 70-30 split for tactics-technical individual, both exercises are usually merged to mimic game situations. Finally, the players solicitations regarding nutrition and hydration purposes were made on a personal basis.

For future studies, it would be great to be able to relate the study's conclusions to contextual factors, such as: what happens when the team plays at home or away, what happens when the team loses or wins, what happens when the team is playing with one more player or less player, etc.

Response: We added it to our conclusion, thank you for your feedback.

---

## [Decision Letter · Decision Letter 1]

25 Jul 2023

A deep dive into the use of Local Positioning System in professional handball: Automatic detection of players’ orientation, position and game phases to analyse specific physical demands

PONE-D-23-10764R1

Dear Dr. Guignard,

We’re pleased to inform you that your manuscript has been judged scientifically suitable for publication and will be formally accepted for publication once it meets all outstanding technical requirements.

Kind regards,

Rabiu Muazu Musa, PhD

Academic Editor

PLOS ONE

Additional Editor Comments (optional):

Reviewers' comments:

Reviewer's Responses to Questions

**Comments to the Author**

1. If the authors have adequately addressed your comments raised in a previous round of review and you feel that this manuscript is now acceptable for publication, you may indicate that here to bypass the “Comments to the Author” section, enter your conflict of interest statement in the “Confidential to Editor” section, and submit your "Accept" recommendation.

Reviewer #1: All comments have been addressed

2. Is the manuscript technically sound, and do the data support the conclusions?

Reviewer #1: Yes

3. Has the statistical analysis been performed appropriately and rigorously? 

Reviewer #1: Yes

4. Have the authors made all data underlying the findings in their manuscript fully available?

Reviewer #1: Yes

5. Is the manuscript presented in an intelligible fashion and written in standard English?

Reviewer #1: Yes

6. Review Comments to the Author

Reviewer #1: (No Response)

7. PLOS authors have the option to publish the peer review history of their article (what does this mean?). If published, this will include your full peer review and any attached files.

Reviewer #1: **Yes: **Demetrio Lozano Jarque

<quillbot-extension-portal></quillbot-extension-portal>

---

## [Editor Report · Acceptance letter]

8 Aug 2023

PONE-D-23-10764R1 

Automatic detection of players’ orientation, position and game phases to analyse specific physical demands 

Dear Dr. Guignard:

I'm pleased to inform you that your manuscript has been deemed suitable for publication in PLOS ONE. Congratulations! Your manuscript is now with our production department. 

Kind regards, 

on behalf of

Dr. Rabiu Muazu Musa 

Academic Editor

PLOS ONE